# Classification of Vitamin D Status Based on Vitamin D Metabolism: A Randomized Controlled Trial in Hypertensive Patients

**DOI:** 10.3390/nu16060839

**Published:** 2024-03-14

**Authors:** Sieglinde Zelzer, Andreas Meinitzer, Dietmar Enko, Martin H. Keppel, Markus Herrmann, Verena Theiler-Schwetz, Christian Trummer, Lisa Schmitt, Andreas Tomaschitz, Patrick Sadoghi, Jutta Dierkes, Pawel Pludowski, Armin Zittermann, Winfried März, Stefan Pilz

**Affiliations:** 1Clinical Institute of Medical and Chemical Laboratory Diagnostics, Medical University of Graz, 8036 Graz, Austria; sieglinde.zelzer@medunigraz.at (S.Z.); andreas.meinitzer@medunigraz.at (A.M.); enko.dietmar@gmx.at (D.E.); keppel.martin@gmail.com (M.H.K.); markus.herrmann@medunigraz.at (M.H.); winfried.maerz@synlab.de (W.M.); 2Institute of Clinical Chemistry and Laboratory Medicine, Hospital Hochsteiermark, Vordernberger Straße 42, 8700 Leoben, Austria; 3Department of Internal Medicine, Division of Endocrinology and Diabetology, Medical University of Graz, Auenbruggerplatz 15, 8036 Graz, Austria; verena.schwetz@medunigraz.at (V.T.-S.); christian.trummer@medunigraz.at (C.T.); l.schmitt@medunigraz.at (L.S.); 4Health Center Trofaiach-Gössgrabenstrasse, 8793 Trofaiach, Austria; andreas.tomaschitz@gmx.at; 5Department of Orthopedics and Trauma, Medical University of Graz, Auenbruggerplatz 5, 8036 Graz, Austria; patrick.sadoghi@medunigraz.at; 6Mohn Nutrition Research Laboratory, Centre for Nutrition, Department of Clinical Medicine, University of Bergen, 5020 Bergen, Norway; jutta.dierkes@uib.no; 7Department of Biochemistry, Radioimmunology and Experimental Medicine, The Children’s Memorial Health Institute, 04-730 Warsaw, Poland; p.pludowski@ipczd.pl; 8Clinic for Thoracic and Cardiovascular Surgery, Herz- und Diabeteszentrum Nordrhein-Westfalen (NRW), Ruhr University Bochum, 32545 Bad Oeynhausen, Germany; azittermann@hdz-nrw.de; 9SYNLAB Academy, Synlab Holding Deutschland GmbH, 68159 Mannheim, Germany; 10Vth Department of Medicine (Nephrology, Hypertensiology, Rheumatology, Endocrinology, Diabetology, Lipidology), Medical Faculty Mannheim, University of Heidelberg, 68167 Mannheim, Germany

**Keywords:** vitamin D, vitamin D metabolite ratio, RCT, supplementation, deficiency, 24,25-dihydroxyvitamin D, classification

## Abstract

Circulating 25-hydroxyvitamin D (25(OH)D) is the generally accepted indicator of vitamin D status. Since hydroxylation of 25(OH)D to 24-25-dihydroxyvitamin D (24,25(OH)2D) is the first step of its catabolism, it has been suggested that a low 24,25(OH)D level and a low vitamin D metabolite ratio (VMR), i.e., 24,25(OH)2D divided by 25(OH)D, may indicate high vitamin D requirements and provide additional diagnostic information beyond serum 25(OH)D. We, therefore, evaluated whether the classification of “functional vitamin D deficiency”, i.e., 25(OH)D below 50 nmol/L, 24,25(OH)2D below 3 nmol/L and a VMR of less than 4%, identifies individuals who benefit from vitamin D supplementation. In participants of the Styrian Vitamin D Hypertension trial, a randomized controlled trial (RCT) in 200 hypertensive patients with serum 25(OH)D below 75 nmol/L, who received either 2.800 international units of vitamin D per day or placebo over 8 weeks, 51 participants had functional vitamin D deficiency. In these individuals, there was no treatment effect of vitamin D supplementation on various parameters of bone metabolism and cardiovascular risk except for a significant effect on parathyroid hormone (PTH) and expected changes in vitamin D metabolites. In conclusion, a low vitamin D metabolite profile did not identify individuals who significantly benefit from vitamin D supplementation with regard to bone markers and cardiovascular risk factors. The clinical significance of functional vitamin D deficiency requires further evaluation in large vitamin D RCTs.

## 1. Introduction

There exists a wide consensus that the assessment of vitamin D status is performed by measuring serum 25-hydroxyvitamin D (25(OH)D) concentrations [1,2]. An ongoing scientific controversy prevails regarding the precise 25(OH)D cut-offs for defining vitamin D deficiency and sufficiency [1,3,4]. Due to unknown reasons, vitamin D requirements seem to be met at extremely low serum 25(OH)D concentrations for some individuals, whereas others require higher serum 25(OH)D levels to satisfy their vitamin D needs [3,5,6,7,8]. Measuring 24,25-dihydroxyvitamin D (24,25(OH)2D) in addition to 25(OH)D and calculating the vitamin D metabolite ratio (VMR) out of these two molecules (i.e., 24,25(OH)2D divided by 25(OH)D) may serve as an indicator of “functional vitamin D deficiency” that may overcome some of the limitations of exclusively measuring serum 25(OH)D for vitamin D status assessment [9]. The basic concept behind this approach is that the hydroxylation of 25(OH)D to 24,25(OH)2D is considered the first step in the catabolism of vitamin D. Assuming that the human body regulates, at least in part, vitamin D metabolism according to the individual demands, a lower catabolism, and thus lower relative concentration of 24,25(OH)2D and a subsequently lower VMR, may hypothetically indicate a higher vitamin D requirement and vice versa [9] (see Figure 1).

There is still no established consensus regarding a universal definition of functional vitamin D deficiency, but based on previous reports, we classify functional vitamin D deficiency in individuals with 25(OH)D below 50 nmol/L (multiplied by 2.496 to convert ng/mL to nmol/L), 24,25(OH)2D below 3 nmol/L and a VMR of less than 4% [9].

Controversial data exist on whether VMR predicts increases in 25(OH)D after vitamin D supplementation, with a previous investigation of our group failing to document such an effect [10]. By contrast, some studies have already suggested that assessment of vitamin D status by using the VMR for defining functional vitamin D deficiency may provide useful information in addition to serum 25(OH)D alone [9,11,12,13,14,15]. In this context, better characterization of bone health and mortality risk has been reported for this approach as opposed to conventional vitamin D status assessment [9,13,15]. Data on this issue are, however, limited to only a few studies, and there exists a knowledge gap regarding randomized controlled trials (RCTs) evaluating clinically relevant vitamin D effects in persons with functional vitamin D deficiency.

In this work, we aim to evaluate the hypothesis that the definition of functional vitamin D deficiency is able to identify individuals who significantly benefit from vitamin D supplementation. This study is a post hoc analysis of the Styrian Vitamin D Hypertension Study, an RCT in 200 hypertensive patients with low 25(OH)D, on vitamin D supplementation with 2.800 international units (IU) (50 µg) daily for 8 weeks [16]. In individuals with serum 25(OH)D concentrations below 50 nmol/L, we evaluate, as our primary outcome analyses, whether those with functional vitamin D deficiency significantly benefit from vitamin D supplementation concerning markers of bone metabolism and cardiovascular risk. In addition, we evaluate in cross-sectional analyses whether individuals with and without functional vitamin D deficiency differ with regard to various clinical and laboratory characteristics.

## 2. Materials and Methods

### 2.1. Study Design

The Styrian Vitamin D Hypertension Study is a double-blind, placebo-controlled trial in 200 hypertensive patients with serum 25(OH)D below 75 nmol/L from the Department of Internal Medicine of the Medical University of Graz. In total, 518 study participants were screened for this study (of whom 200 were included in the RCT) and are thus part of the entire study cohort termed Styrian Hypertension Study. The main objective of the initial trial was to evaluate whether vitamin D supplementation with 2.800 international units (IU) daily for 8 weeks reduces 24 h systolic ambulatory blood pressure (ABP) compared to placebo. Secondary endpoints included 24 h diastolic ABP and other cardiovascular risk factors. All study participants gave written informed consent, and the ethics committee at the Medical University of Graz, Austria, approved the initial study and the additional analyses of this present post hoc analysis. The study was registered at EU Clinical Trials Register (http://www.clinicaltrialsregister.eu (accessed on 23 February 2024), EudraCT number 2009-018125-70) and at clinicaltrials.gov (ClinicalTrials.gov Identifier NCT02136771). The publications of this RCT adhere to the Consolidated Standards of Reporting Trials (CONSORT) 2010 statement. Further details on the study methods and the main study findings of this RCT were published elsewhere [16]. For the current investigation, we included only individuals with available data to calculate the VMR, i.e., data on 25(OH)D and 24,25(OH)2D.

### 2.2. Laboratory Measurements

In serum samples that were stored at −80° Celsius from blood collection until analysis in October 2023, 25(OH)D and 24,25(OH)2D were determined by a validated LC-MS/MS method that is regularly evaluated by internal and external quality controls with satisfactory results [17]. For external quality assessment, this method is enrolled in the Vitamin D External Quality Assessment Scheme (DEQAS) that uses target values assigned by the Centers for Disease Control and Prevention (CDC) reference measurement procedure (i.e., certified reference material (SRM 2972a) from the National Institute of Standards and Technology (NIST)). For all vitamin D metabolites, the intra- and inter-day imprecision was <9% and <12%, respectively, and the limit of detection (LOD) and limit of quantification (LOQ) ranged from 0.12 to 0.60 ng/mL (0.3 to 1.5 nmol/L) and 0.40 to 1.24 ng/mL (1.0 to 3.1 nmol/L), respectively. Recovery varied between 76.1% and 84.3%. Regarding bone markers, β-CrossLaps (CTX), osteocalcin and procollagen type 1 amino-terminal propetide (P1NP) were determined by electrochemiluminescence immunoassays (Roche Diagnostics, Mannheim, Germany) according to manufacturers’ instructions. Bone-specific alkaline phosphatase (bALP) was measured by a spectrophotometric immunoassay (Immunodiagnostic Systems Ltd., Boldon, UK). 1,25-dihydroxyvitamin D (1,25(OH)2D) concentrations were determined by a chemoluminescence immunoassay (Immunodiagnostic Systems Ltd., Boldon, UK) and *C*-terminal fibroblast growth factor-23 (FGF-23) by a multi matrix ELISA (BIOMEDICA Medizinprodukte GmbH & CO KG, Vienna, Austria). In addition to vitamin D metabolites and bone markers, several other laboratory parameters were determined by methods that have been described elsewhere [16].

### 2.3. Outcome Measures

All primary and secondary outcome measures of the initial RCT were re-analyzed as part of this investigation. In addition, for analyses on bone and mineral metabolism, we also included 24,25(OH)2D, VMR, 1,25(OH)2D, FGF-23, bALP, CTX, osteocalcin and P1NP.

### 2.4. Statistical Analyses

Continuous data with a normal distribution are shown as means with standard deviation (SD), and non-normally distributed variables are shown as medians with 25th to 75th percentile. Categorical variables are presented as percentages. Non-normally distributed variables are log(e)-transformed before use in parametric procedures. Group comparisons of individuals with and without functional vitamin D deficiency are performed by Chi-Square test and Student’s *t* test. In individuals with functional vitamin D deficiency, analysis of co-variance (ANCOVA) with adjustment for baseline values is used to test for differences between the vitamin D and the placebo group. No data imputation was performed for missing values. All analyses are exploratory for these post hoc analyses. A power calculation revealed that the expected sample sizes for our subgroup analyses in participants with functional vitamin D deficiency seem reasonable for most outcomes studied. For example, for the primary outcome measure of the initial RCT publication, i.e., 24 h systolic ABP, we calculated, with a power of 80% and an α of 0.05, a sample size of 22 participants per study group (44 participants overall) when assuming an effect size of 8 mm Hg and an SD of 9 mm Hg. A *p*-value below 0.05 was considered statistically significant. Statistical analyses were performed by using SPSS Version 27 (SPS, Chicago, IL, USA).

## 3. Results

Data on the VMR were available in 505 out of 518 persons who were screened for this study, of whom 192 suffered from vitamin D deficiency characterized by a serum 25(OH)D concentration below 50 nmol/L. In Table 1, we present the baseline characteristics of all persons screened for this study with available LC-/MS-MS-based measurements of vitamin D metabolites, stratified into those with 25(OH)D below 50 nmol/L and ≥50 nmol/L, respectively. 

In Table 2, we present data for all participants screened for this study with 25(OH)D below 50 nmol/L, comparing baseline characteristics of groups with versus without functional vitamin D deficiency.

In Table 3, we present parameters of mineral metabolism at baseline and follow-up, as well as the treatment effect of vitamin D in study participants of the RCT with serum 25(OH)D below 50 nmol/L and functional vitamin D deficiency. 

In Table 4, we present cardiovascular risk factors at baseline and follow-up, and the treatment effect of vitamin D in study participants of the RCT with serum 25(OH)D below 50 nmol/L and functional vitamin D deficiency.

Our results remained materially unchanged when males and females were analyzed separately.

## 4. Discussion

In vitamin D-deficient hypertensive patients, those with functional vitamin D deficiency did not yield significant benefits in response to vitamin D supplementation with regard to markers of bone health and cardiovascular risk except for a decrease in PTH concentrations and expected changes in vitamin D metabolites. Cross-sectional analyses in vitamin D-deficient individuals showed, amongst others, a significantly higher prevalence of diabetes mellitus and disturbed glucose metabolism in those with versus without functional vitamin D deficiency.

Our study is, to the best of our knowledge, the first to evaluate whether individuals with “functional vitamin D deficiency” significantly benefit from vitamin D supplementation regarding markers of bone health and cardiovascular risk. As in our initial publication of this RCT, we did not detect significant benefits of vitamin D supplementation apart from well-established effects on PTH and expected changes in vitamin D metabolites. Our work was stimulated by previous observational data suggesting that individuals with functional vitamin D deficiency, i.e., those with relatively low 24,25(OH)2D concentrations and a low VMR, had adverse bone health and higher mortality risk as compared to individuals with an unremarkable vitamin D profile [9,13,15]. In this context, we could not confirm the findings derived from other cohorts that individuals with functional vitamin D deficiency had an adverse profile regarding bone turnover markers [9]. These inconsistent findings may be due to different cohort characteristics, a limited sample size of our study as compared to previous investigations, or other, yet unknown, reasons. Although vitamin D supplementation did not alter cardiovascular risk markers in our RCT analyses, we observed that the prevalence of diabetes mellitus and impaired glucose metabolism was significantly increased in individuals with versus without functional vitamin D deficiency. This is an interesting finding that warrants further investigations that specifically address the relationship between functional vitamin D deficiency and diabetes mellitus and evaluate whether the risk of diabetes mellitus and disturbed glucose metabolism can be significantly reduced by vitamin D supplementation in individuals suffering from functional vitamin D deficiency. In addition to lower 1,25(OH)2D and other vitamin D metabolites, older age and a lower GFR were also more prevalent in individuals with functional vitamin D deficiency, and one might, therefore, speculate that the lower vitamin D catabolism associated with these parameters may hypothetically indicate a higher vitamin D requirement, unconsidered confounding factors or alterations in vitamin D metabolism related to these parameters.

Despite our findings of no significant benefits of vitamin D supplementation on bone metabolism and cardiovascular risk markers, we continue to urge other scientists to further evaluate whether the concept of “functional vitamin D deficiency” is useful for a better characterization and classification of vitamin D status. Many large vitamin D RCTs have been performed and recently published reporting, by the vast majority, no significant vitamin D effects, probably due to enrolling individuals with largely sufficient serum 25(OH)D concentrations. Measuring 24,25(OH)2D, if not yet available, in these RCTs and re-analyzing the existing data with analyses restricted to participants with functional vitamin D deficiency could be considered a great opportunity to further evaluate our hypothesis on whether individuals with functional vitamin D deficiency are particularly sensitive to beneficial vitamin D effects. Importantly, some vitamin D RCTs with no significant effect in the overall, largely vitamin D-sufficient study population have shown benefits of vitamin D when restricting their analyses to individuals with very low 25(OH)D concentrations [18,19,20,21]. As our observational data suggest a particularly high risk of disturbed glucose metabolism in participants with functional vitamin D deficiency, we would particularly recommend evaluating the concept of functional vitamin D deficiency in vitamin D RCTs on diabetes-related outcomes [18,21,22]. 

While our findings do not support the hypothesis that individuals with functional vitamin D deficiency significantly benefit from vitamin D supplementation, we wish to underline that our LC-MS/MS method for simultaneous measurements of different vitamin D metabolites has some advantages over the widely used immuno-assay based determinations of serum 25(OH)D. First, despite significant efforts for quality control and standardization, a significant variability remains in the analytical performance of immuno-assays as opposed to LC/MS-MS methods [23,24]. Second, simultaneous measurements of 25(OH)D and 24,25(OH)2D by LC-MS/MS-based methods allow for the identification of individuals suffering from 24-hydroxylase deficiency caused by pathogenic mutations of CYP24A1 [25]. This inherited disease is characterized by hypercalcemia due to an impaired catabolism of vitamin D metabolites, with thus a low 24,25(OH)2D-to-25(OH)D ratio [25]. In addition, routine use of simultaneous measurement of 25(OH)D and 24,25(OH)2D by LC-MS/MS methods is also useful to distinguish between vitamin D intoxication and 24-hydroxylase deficiency due to pathogenic CYP24A1 mutations, as those with vitamin D intoxication have significantly higher 24,25(OH)2D concentrations when compared to those suffering from 24-hydroxylase deficiency [11]. We also want to stress that we are well aware that our assumption of functional vitamin D deficiency as a condition indicating high vitamin D requirements based on the consideration that the human body reduces vitamin D catabolism reflected by low 24,25(OH)D levels according to the prevailing vitamin D demand is just a simplified hypothesis. Therefore, we stress that there exists a complex and partially unknown regulation of vitamin D metabolism, in particular of 24-hydroxylase activity, with many interacting factors [26,27,28]. Further research is warranted to better characterize vitamin D metabolism with the aim to further improve the identification of individuals who are most likely to benefit from vitamin D supplementation and to critically evaluate and eventually optimize the classification of functional vitamin D deficiency as applied in our study. 

We have to acknowledge that our study is only a post hoc analysis of an RCT, and as only a small proportion of our trial participants suffered from functional vitamin D deficiency, we had a limited sample size. Nevertheless, applying a formal power calculation reveals that the expected sample sizes for our subgroup analyses in participants with functional vitamin D deficiency seem reasonable for most outcomes studied to detect clinically meaningful treatment effects (in the range of slightly less than one SD of the outcome measure) with a power of 80%. External validity (generalizability) may also be limited as we investigated a cohort with arterial hypertension. Moreover, the study duration may have been too short to observe some effects that require long-term vitamin D supplementation. Other limitations are the relatively low number of participants with very low 25(OH)D levels (e.g., below 25/30 nmol/L) and missing data on vitamin D status before study inclusion. While our results argue against relatively large treatment effects of vitamin D supplementation in patients with functionally vitamin D deficiency, larger RCTs are required to evaluate whether there are smaller but still relevant treatment effects on a population level. Regarding the classification of functional vitamin D deficiency, we adhered to a previously published empirical definition for this but are well aware that other approaches for this classification might be superior [9]. Our cross-sectional analyses and group comparisons are limited because they are only based on crude (unadjusted) analyses. We refrained from conducting multivariate-adjusted analyses regarding these cross-sectional data as the main aim of this work was to focus on the effect of vitamin D intervention rather than on association analyses. In this context, we emphasize the need to test our hypothesis on functional vitamin D deficiency as a tool to identify individuals with high sensitivity to vitamin D supplementation in RCTs rather than performing multiple observational studies on this topic. Therefore, the main strengths of our work are that we tested our hypothesis in an RCT and followed an a priori-specified statistical analysis plan.

## 5. Conclusions

In conclusion, our criteria for functional vitamin D deficiency did not show that vitamin D-deficient patients significantly benefit from vitamin D supplementation regarding bone markers and cardiovascular risk factors, except for a decrease in PTH and expected changes in vitamin D metabolites. Further studies, in particular large RCTs, are warranted to explore whether the measurements of vitamin D metabolites in addition to 25(OH)D are useful for the classification of vitamin D status and identification of individuals who particularly benefit from vitamin D treatment.

## Figures and Tables

**Figure 1 nutrients-16-00839-f001:**
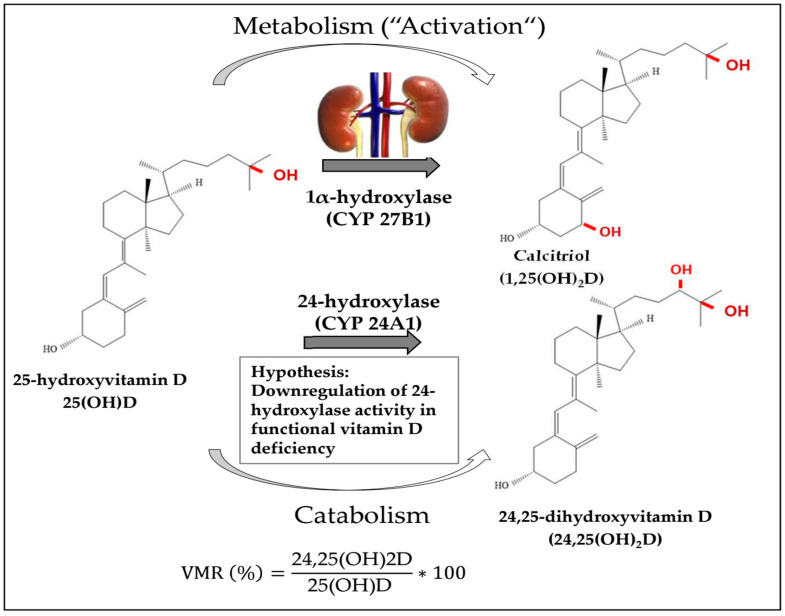
This figure shows that 25(OH)D is “activated” by 1α-hydroxylase to calcitriol, whereas 24-hydroxylase converts 25(OH)D to 24,25(OH)2D as the first step in vitamin D catabolism. We hypothesize that the catabolism by 24-hydroxylase is reduced in individuals with high vitamin D requirements, e.g., in those with functional vitamin D deficiency.

**Table 1 nutrients-16-00839-t001:** Baseline characteristics of all study participants, stratified according to serum 25-hydroxyvitamin D in those <50 nmol/L and ≥50 nmol/L.

Characteristics	All (*n* = 505)	25(OH)D < 50 nmol/L (*n* = 195)	25(OH)D ≥ 50 nmol/L (*n* = 310)	*p* Value
Females (%)	52.3	50.7	53.2	0.591
Age (years)	61.2 ± 10.6	59.9 ± 11.3	62.0 ± 10.0	0.026
Body mass index (kg/m^2^)	29.6 ± 5.1	30.8 ± 5.7	28.9 ± 4.6	<0.001
Office systolic BP (mm Hg)	141 ± 16.6	141 ± 17.1	141 ± 16.3	0.734
Office diastolic BP (mm Hg)	86.3 ± 11.0	86.7 ± 11.7	86.1 ± 10.5	0.573
24 h systolic BP (mm Hg)	128 ± 13.9	130 ± 14.3	126 ± 13.4	0.002
24 h diastolic BP (mm Hg)	76.4 ± 9.5	77.5 ± 10.3	75.8 ± 9.0	0.049
NT-proBNP (pg/mL)	82 (42–153)	83 (40.8–150)	84 (45–158)	0.428
Corrected QT interval (ms)	413 (391–441)	414 (391–445)	413 (391–439)	0.865
PRC (µU/mL)	17.4 (10.1–48.3)	18.6 (10.2–57.5)	16.9 (9.6–44.3)	0.596
PAC (ng/dL)	17.0 ± 10.3	16.5 ± 10.2	17.3 ± 10.4	0.39
eGFR (mL/min/1.73 m^2^)	79.2 ± 17.7	81.8 ± 18.7	78.1 ± 16.9	0.021
24 h UAE (mg/24 h)	8.5 (6.0–19.8)	9.0 (6.0–21.0)	8.0 (6.0–17.0)	0.064
Diabetes mellitus (%)	26.5	34.9	21.3	<0.001
Fasting glucose (mg/dL)	112 ± 40.8	121 ± 49.8	107 ± 33.0	<0.001
HbA1c (mmol/mol)	43.6 ± 11.7	46.2 ± 14.2	41.9 ± 9.4	<0.001
HOMA-IR	1.73 (1.03–3.06)	2.01 (1.24–3.93)	1.56 (0.97–2.76)	0.005
Triglycerides (mg/dL)	110 (75.0–154)	117 (73.8–164)	105 (75.0–151)	0.311
HDL cholesterol (mg/dL)	58.6 ± 17.3	57.1 ± 17.4	59.5 ± 17.2	0.142
LDL cholesterol (mg/dL)	114 ± 38.4	113 ± 37.9	115 ± 38.7	0.470
PWV (m/s)	8.6 ± 2.4	8.5 ± 2.0	8.6 ± 2.6	0.861
CRP (mg/L)	1.80 (0.80–3.40)	1.70 (0.85–3.40)	1.9 (0.80–3.50)	0.933
25(OH)D (nmol/L)	60.4 ± 26.7	36.1 ± 9.6	75.7 ± 22.4	<0.001
24,25(OH)2D (nmol/L)	3.7 ± 2.6	1.9 ± 1.1	5.1 ± 2.4	<0.001
VMR (%)	6.2 ± 2.3	5.2 ± 2.4	6.8 ± 2.0	<0.001
1,25(OH)2D (pg/mL)	52.5 ± 19.8	44.9 ± 17.4	57.2 ± 19.7	<0.001
Fibroblast growth factor-23 (pmol/L)	0.85 (0.60–1.25)	0.88 (0.61–1.37)	0.83 (0.59–1.21)	0.072
bALP (µg/L)	16.9 ± 6.2	17.3 ± 6.9	16.6 ± 5.7	0.380
CTX (ng/mL)	0.22 ± 0.18	0.20 ± 0.15	0.24 ± 0.20	0.058
Osteocalcin (ng/mL)	14.9 ± 8.4	14.4 ± 8.2	15.3 ± 8.5	0.216
P1NP (ng/mL)	42.0 ± 21.0	40.7 ± 18.3	42.8 ± 22.5	0.283
PTH (pg/mL)	51.1 ± 19.5	56.8 ± 22.0	47.4 ± 16.7	<0.001
Plasma calcium (mmol/L)	2.38 ± 0.11	2.37 ± 0.11	2.39 ± 0.10	0.017
24 h UCaE (mmol/24 h)	3.72 ± 2.48	3.59 ± 2.62	3.80 ± 2.39	0.397
Calcium supplement (%)	9.9	5.1	12.9	0.005
Vitamin D supplement (%)	10.9	5.1	14.5	0.001

Data are presented as means with standard deviation, medians with interquartile range or as percentages. Comparisons between the vitamin D and placebo group were calculated with Student’s t test or with Chi-Square test. BP, blood pressure; NT-proBNP, *N*-terminal pro-B-type natriuretic peptide; PRC, plasma renin concentration; PAC, plasma aldosterone concentration; eGFR, estimated Glomerular Filtration Rate; UAE, Urinary Albumin Excretion; HOMA-IR, Homeostatis Model Assessment—Insulin Resistance; PWV, pulse wave velocity; CRP, C-Reactive Protein; 25(OH)D, 25-hydroxyvitamin D; 24,25(OH)2D, 24,25-dihydroxyvitamin D; VMR, vitamin D metabolite ratio; 1,25(OH)2D, 1,25-dihydroxyvitamin D, bALP; bone-specific alkaline phosphatase; CTX, β-CrossLaps; P1NP, procollagen type 1 amino-terminal propeptide; PTH, parathyroid hormone; UCaE, urinary calcium excretion.

**Table 2 nutrients-16-00839-t002:** Baseline characteristics of study participants with serum 25-hydroxyvitamin D < 50 nmol/L stratified in patients with and without functional vitamin D deficiency *.

Characteristics	25(OH)D < 50 nmol/L (*n* = 66) with Functional Deficiency	25(OH)D < 50 nmol/L (*n* = 126) without Functional Deficiency	*p* Value
Females (%)	50.0	51.2	0.741
Age (years)	62.3 ± 10.2	58.7 ± 11.7	0.035
Body mass index (kg/m^2^)	31.0 ± 5.8	30.7 ± 5.6	0.722
Office systolic BP (mm Hg)	144 ± 18.1	140 ± 16.6	0.187
Office diastolic BP (mm Hg)	86.4 ± 11.6	86.9 ± 11.8	0.780
24 h systolic BP (mm Hg)	131 ± 15.8	130 ± 13.5	0.550
24 h diastolic BP (mm Hg)	76.7 ± 9.6	77.9 ± 10.6	0.470
NT-proBNP (pg/mL)	82.5 (39.0–171)	87.5 (37.3–165)	0.717
Corrected QT interval (ms)	407 (388–437)	416 (392–442)	0.150
PRC (µU/mL)	19.0 (12.4–79.3)	17.4 (8.5–51.8)	0.127
PAC (ng/dL)	18.5 ± 9.4	15.5 ± 10.6	0.055
eGFR (mL/min/1.73 m^2^)	76.5 ± 21.1	84.6 ± 16.8	0.004
24 h UAE (mg/24 h)	9.0 (6.0–24.5)	9.0 (6.0–21.0)	0.604
Diabetes mellitus (%)	54.4	45.6	<0.001
Fasting glucose (mg/dL)	141 ± 61.2	110 ± 39.0	<0.001
HbA1c (mmol/mol)	52.6 ± 17.5	42.8 ± 10.8	<0.001
HOMA-IR	2.6 (1.5–5.7)	2.1 (1.3–3.5)	0.061
Triglycerides (mg/dL)	117 (70.5–178)	118 (81.8–150)	0.504
HDL cholesterol (mg/dL)	55.8 ± 20.3	57.8 ± 15.7	0.437
LDL cholesterol (mg/dL)	107 ± 38.7	116 ± 37.2	0.115
PWV (m/s)	8.8 ± 2.0	8.4 ± 2.1	0.240
CRP (mg/L)	1.95 (1.10–4.75)	1.90 (0.90–3.40)	0.017
25(OH)D (nmol/L)	31.5 ± 9.7	38.4 ± 8.8	<0.001
24,25(OH)2D (nmol/L)	0.85 ± 0.46	2.38 ± 0.96	<0.001
VMR (%)	2.76 ± 0.93	6.43 ± 1.85	<0.001
1,25(OH)2D (pg/mL)	39.4 ± 18.0	47.7 ± 16.4	0.001
Fibroblast growth factor-23 (pmol/L)	1.00 (0.69–1.44)	0.89 (0.57–1.37)	0.383
bALP (µg/L)	18.1 ± 7.2	17.0 ± 6.7	0.556
CTX (ng/mL)	0.19 ± 0.14	0.21 ± 0.15	0.454
Osteocalcin (ng/mL)	14.0 ± 9.4	14.5 ± 7.5	0.681
P1NP (ng/mL)	37.4 ± 14.4	42.3 ± 19.8	0.083
PTH (pg/mL)	57.1 ± 26.7	56.7 ± 19.3	0.917
Plasma calcium (mmol/L)	2.37 ± 0.11	2.37 ± 0.11	0.958
24 h UCaE (mmol/24 h)	3.44 ± 2.79	3.67 ± 2.53	0.592
Calcium supplement (%)	2.0	6.3	0.499
Vitamin D supplement (%)	4.5	5.6	0.900

* Functional vitamin D deficiency is classified in persons with 24,25(OH)2D below 3 nmol/L and a VMR of less than 4%. Data are presented as means with standard deviation, medians with interquartile range or as percentages. Comparisons between the vitamin D and placebo group were calculated with Student’s *t* test or with Chi-Square test. BP, blood pressure; NT-proBNP, *N*-terminal pro-B-type natriuretic peptide; PRC, plasma renin concentration; PAC, plasma aldosterone concentration; eGFR, estimated Glomerular Filtration Rate; UAE, Urinary Albumin Excretion; HOMA-IR, Homeostatis Model Assessment—Insulin Resistance; PWV, pulse wave velocity; CRP, C-Reactive Protein; 25(OH)D, 25-hydroxyvitamin D; 24,25(OH)2D, 24,25-dihydroxyvitamin D; VMR, vitamin D metabolite ratio; 1,25(OH)2D, 1,25-dihydroxyvitamin D; bALP, bone-specific alkaline phosphatase; CTX, β-CrossLaps; P1NP, procollagen type 1 amino-terminal propeptide; PTH, parathyroid hormone; UCaE, urinary calcium excretion.

**Table 3 nutrients-16-00839-t003:** Parameters of mineral metabolism at baseline and follow-up, as well as changes from baseline in study participants with functional vitamin D deficiency and available values at both study visits.

Characteristics	Baseline	Follow-Up	Treatment Effect	*p* Value
25-hydroxyvitamin D (nmol/L)				
Vitamin D (*n* = 27)	38.9 ± 14.3	68.8 ± 11.2	30.9 (24.4 to 37.5)	<0.001
Placebo (*n* = 22)	38.9 ± 12.9	37.1 ± 14.0		
24,25(OH)2D (nmol/L)				
Vitamin D (*n* = 25)	1.16 ± 0.62	4.54 ± 1.14	2.6 (2.0 to 3.3)	<0.001
Placebo (*n* = 22)	1.15 ± 0.57	1.88 ± 1.11		
Vitamin D metabolite ratio (%)				
Vitamin D (*n* = 25)	3.00 ± 0.95	6.79 ± 1.67	1.9 (1.1 to 2.6)	<0.001
Placebo (*n* = 22)	3.00 ± 0.67	5.04 ± 1.59		
1,25-dihydroxyvitamin D (pg/mL)				
Vitamin D (*n* = 19)	50.5 ± 20.9	61.8 ± 27.6	6.2 (−7.8 to 20.1)	0.377
Placebo (*n* = 18)	38.3 ± 17.7	43.3 ± 17.2		
Fibroblast growth factor-23 (pmol/L)				
Vitamin D (*n* = 18)	0.83 ± 0.71	2.01 ± 4.35	1.5 (−0.56 to 3.61)	0.147
Placebo (*n* = 17)	1.34 ± 1.30	1.14 ± 0.49		
Bone-specific alkaline phosphatase (µg/L)				
Vitamin D (*n* = 17)	16.5 ± 7.6	16.6 ± 5.7	0.67 (−2.95 to 1.61)	0.553
Placebo (*n* = 16)	19.8 ± 6.4	19.2 ± 6.5		
β-CrossLaps (ng/mL)				
Vitamin D (*n* = 14)	0.19 ± 0.12	0.18 ± 0.11	−0.011 (−0.078 to 0.057)	0.747
Placebo (*n* = 16)	0.19 ± 0.14	0.23 ± 0.16		
Osteocalcin (ng/mL)				
Vitamin D (*n* = 18)	12.6 ± 6.9	13.2 ± 5.9	0.46 (−2.060 to 2.986)	0.711
Placebo (*n* = 17)	13.4 ± 6.7	14.4 ± 5.3		
Procollagen typ 1 amino-terminal propetide (ng/mL)				
Vitamin D (*n* = 18)	35.6 ± 17.5	38.5 ± 16.6	−1.435 (−6.52 to 9.39)	0.715
Placebo (*n* = 17)	40.2 ± 13.9	41.1 ± 11.7		
Parathyroid hormone (pg/mL)#				
Vitamin D (*n* = 19)	52.2 ± 23.4	46.9 ± 15.7	−12.2 (−22.1 to −2.3)	0.017
Placebo (*n* = 20)	50.9 ± 21.6	55.4 ± 29.6		
Plasma calcium (mmol/L)				
Vitamin D (*n* = 19)	2.38 ± 0.08	2.36 ± 0.07	−0.008 (−0.055 to 0.040)	0.747
Placebo (*n* = 20)	2.38 ± 0.14	2.37 ± 0.13		
24 h urinary calcium excretion (mmol/24 h) #				
Vitamin D (*n* = 12)	3.30 (2.10–7.20)	3.74 (2.05–6.86)	−0.19 (−1.34 to 1.72)	0.802
Placebo (*n* = 15)	2.05 (1.28–5.40)	2.53 (0.65–6.24)		

Data at baseline and follow-up are shown as means with standard deviation or as medians with interquartile range. Treatment effects (with 95% confidence intervals) and *p* values were calculated by analysis of co-variance (ANCOVA) for group differences at follow-up with adjustment for baseline values; # skewed variable for which logarithmically transformed values were used in ANCOVA, but untransformed values are shown in the table.

**Table 4 nutrients-16-00839-t004:** Cardiovascular outcome variables at baseline and follow-up and changes from baseline in study participants with functional vitamin D deficiency and available values at both study visits.

Characteristics	Baseline	Follow-Up	Treatment Effect	*p* Value
24 h systolic blood pressure (mm Hg)				
Vitamin D (*n* = 24)	131 ± 9.5	129 ± 9.3	−1.14 (−6.10 to 3.83)	0.647
Placebo (*n* = 24)	134 ± 7.4	132 ± 10.6		
24 h diastolic blood pressure (mm Hg)				
Vitamin D (*n* = 24)	78.7 ± 7.6	76.7 ± 8.7	−0.64 (−3.64 to 2.36)	0.668
Placebo (*n* = 24)	77.8 ± 7.3	76.4 ± 8.7		
*N*-terminal pro-B-type natriuretic peptide (ng/L) #				
Vitamin D (*n* = 26)	45.0 (29.5–76.3)	58.5 (35.8–134)	2.55 (−69.5 to 74.6)	0.944
Placebo (*n* = 25)	99.0 (45.5–228)	98.0 (50.5–188)		
Corrected QT interval (ms)				
Vitamin D (*n* = 23)	409 ± 30.8	422 ± 30.8	19.4 (−32.4 to 71.2)	0.455
Placebo (*n* = 24)	417 ± 33.9	423 ± 29.9		
Plasma renin concentration (µU/mL) #				
Vitamin D (*n* = 24)	18.3 (11.4–70.7)	15.0 (10.6–47.0)	−7.78 (−23.5 to 7.99)	0.326
Placebo (*n* = 23)	14.9 (9.0–37.2)	17.1 (8.8–46.7)		
Plasma aldosterone concentration (ng/dL)				
Vitamin D (*n* = 26)	16.8 ± 7.1	17.4 ± 6.6	2.53 (−1.55 to 6.62)	0.218
Placebo (*n* = 25)	19.9 ± 10.9	22.8 ± 14.3		
24 h urinary albumin concentration (mg/24 h) #				
Vitamin D (*n* = 12)	7.0 (6.0–13.0)	8.0 (7.9–11.9)	6.99 (−14.7 to 28.7)	0.509
Placebo (*n* = 14)	8.0 (4.0–120)	8.0 (4.8–23.4)		
Homeostasis Model Assessment—Insulin Resistance #				
Vitamin D (*n* = 26)	2.38 (1.11–5.48)	2.53 (1.38–4.75)	0.41 (−3.1 to 3.9)	0.812
Placebo (*n* = 23)	3.08 (1.40–6.80)	3.79 (1.60–7.28)		
Triglycerides (mg/dL)				
Vitamin D (*n* = 26)	132 ± 78.0	135 ± 80.8	2.63 (−25.2 to 30.5)	0.850
Placebo (*n* = 25)	136 ± 72.4	140 ± 64.7		
HDL cholesterol (mg/dL)				
Vitamin D (*n* = 26)	54.7 ± 15.7	54.9 ± 17.4	0.42 (−3.90 to 4.75)	0.845
Placebo (*n* = 25)	53.2 ± 15.8	53.8 ± 17.2		
Pulse wave velocity (m/s)				
Vitamin D (*n* = 23)	8.68 ± 1.76	8.93 ± 2.32	0.59 (−0.57 to 1.76)	0.308
Placebo (*n* = 15)	9.08 ± 2.58	8.41 ± 2.46		

Data at baseline and follow-up are shown as means with standard deviation or as medians with interquartile range. Treatment effects (with 95% confidence intervals) and *p* values were calculated by analaysis of co-variance (ANCOVA) for group differences at follow-up with adjustment for baseline values; # skewed variable for which logarithmically transformed values were used in ANCOVA but untransformed values are shown in the table.

## Data Availability

The data presented in this study are available on request from the corresponding author. The data are not publicly available due to data protection regulation issues.

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
