# Peer review of "Classification of Vitamin D Status Based on Vitamin D Metabolism: A Randomized Controlled Trial in Hypertensive Patients"

_nutrients, 2024, doi:10.3390/nu16060839_

Round 1
Reviewer 1 Report
Comments and Suggestions for Authors a study with an original approach - calculation of vitamin D metabolite ratio (VMR) out of two molecules (i.e. 24,25(OH)2D divided by 25(OH)D) as an indicator of "functional vitamin D deficiency" to evaluate the hypothesis that the definition of functional vitamin D deficiency is able to identify individuals who significantly benefit from vitamin D supplementation. Although on the whole the authors succeeded, the weak points of the work should be noted: study is only a post-hoc analysis, sample size is limited, the selection of hypertensive patients can be discussed, etc. Thus, further studies, in particular large randomized controlled trials are necessary to explore whether the measurements of vitamin D metabolites in addition to25(OH)D are useful for the classification of vitamin D status. ​
Author Response
Response to the Reviewer
We revised our work accordingly and present a point-by-point reply to the reviewer comments below.
A study with an original approach - calculation of vitamin D metabolite ratio (VMR) out of two molecules (i.e. 24,25(OH)2D divided by 25(OH)D) as an indicator of "functional vitamin D deficiency" to evaluate the hypothesis that the definition of functional vitamin D deficiency is able to identify individuals who significantly benefit from vitamin D supplementation. Although on the whole the authors succeeded, the weak points of the work should be noted: study is only a post-hoc analysis, sample size is limited, the selection of hypertensive patients can be discussed, etc. Thus, further studies, in particular large randomized controlled trials are necessary to explore whether the measurements of vitamin D metabolites in addition to25(OH)D are useful for the classification of vitamin D status.
Response: We totally agree with the reviewer that our study has certain limitations and that further large randomized controlled trials are required to evaluate the utility of measuring vitamin D metabolites for the classification of vitamin D status. According to this comment we have now stressed these limitations mentioned by the reviewer. In detail we included the following sentence in the discussion: “External validity (generalizability) may be limited as we investigated a cohort with arterial hypertension.” Moreover, we have already mentioned the following limitations that were noted by the reviewer: “We have to acknowledge that our study is only a post-hoc analysis of an RCT and as only a small proportion of our trial participants suffered from functional vitamin D deficiency, we had a limited sample size.” Furthermore, we concluded the Abstract with the following sentence: “The clinical significance of functional vitamin D deficiency requires further evaluation in large vitamin D RCTs.” We hope that the reviewer agrees with this.
Reviewer 2 Report
Comments and Suggestions for Authors
I’ve read with attention the paper of Zelzer et al. that is potentially of interest. The background and aim of the study have been clearly defined. The methodology applied is overall correct, the results are reliable and adequately discussed. The references are sufficient in number, adequate and update. The authors have deeply discussed the limitation of their study, however I also think that they should stress the concept that the study was relatively short to observe changes in bone and cardiovascular markers in a so small patient sample without a severe Vit. D deficiency. Moreover, no mention on the duration of the Vit. D deficit is known (and presumably it would be hardly known) and the different duration could also have affected the observed results.
Author Response
Response to the Reviewer
We revised our work accordingly and present a point-by-point reply to the reviewer comments below.
I’ve read with attention the paper of Zelzer et al. that is potentially of interest. The background and aim of the study have been clearly defined. The methodology applied is overall correct, the results are reliable and adequately discussed. The references are sufficient in number, adequate and update.
Response: We thank the reviewer for this positive comment on our manuscript.
The authors have deeply discussed the limitation of their study, however I also think that they should stress the concept that the study was relatively short to observe changes in bone and cardiovascular markers in a so small patient sample without a severe Vit. D deficiency. Moreover, no mention on the duration of the Vit. D deficit is known (and presumably it would be hardly known) and the different duration could also have affected the observed results.
Response: We agree with the reviewer that the short study duration is a limitation as well as the relatively small sample size without many participants suffering from severe vitamin D deficiency. As the reviewer assumes, we do not have data on the duration of vitamin D deficiency before study entry but we agree that this may be of relevance and have therefore mentioned this as a limitation of our investigation. In detail, we included the following sentence in the discussion: “Moreover, the study duration may have been too short to observe some effects that require long-term vitamin D supplementation. Other limitations are the relatively low number of participants with very low 25(OH)D levels (e.g. below 25/30 nmol/L) and missing data on vitamin D status before study inclusion.“ We hope that the reviewer agrees with this.